# Receptor for Advanced Glycation End-Products Promotes Activation of Alveolar Macrophages through the NLRP3 Inflammasome/TXNIP Axis in Acute Lung Injury

**DOI:** 10.3390/ijms231911659

**Published:** 2022-10-01

**Authors:** Woodys Lenga Ma Bonda, Marianne Fournet, Ruoyang Zhai, Jean Lutz, Raiko Blondonnet, Céline Bourgne, Charlotte Leclaire, Cécile Saint-Béat, Camille Theilliere, Corinne Belville, Damien Bouvier, Loïc Blanchon, Marc Berger, Vincent Sapin, Matthieu Jabaudon

**Affiliations:** 1iGReD, Université Clermont Auvergne, CNRS, INSERM, 63000 Clermont-Ferrand, France; 2Department of Perioperative Medicine, CHU Clermont-Ferrand, 63000 Clermont-Ferrand, France; 3Department of Hematology and Immunology, CHU Clermont-Ferrand, 63000 Clermont-Ferrand, France; 4UR 7453 CHELTER, Université Clermont Auvergne, 63000 Clermont-Ferrand, France; 5Department of Medical Biochemistry and Molecular Genetics, CHU Clermont-Ferrand, 63000 Clermont-Ferrand, France

**Keywords:** acute respiratory distress syndrome, acute lung injury, macrophage activation, alveolar inflammation, receptor for advanced glycation end-products, thioredoxin-interacting protein, NLR family, pyrin domain containing 3

## Abstract

The roles of thioredoxin-interacting protein (TXNIP) and receptor for advanced glycation end-products (RAGE)-dependent mechanisms of NOD-like receptor family, pyrin domain containing 3 (NLRP3) inflammasome-driven macrophage activation during acute lung injury are underinvestigated. Cultured THP-1 macrophages were treated with a RAGE agonist (S100A12), with or without a RAGE antagonist; cytokine release and intracytoplasmic production of reactive oxygen species (ROS) were assessed in response to small interfering RNA knockdowns of TXNIP and NLRP3. Lung expressions of TXNIP and NLRP3 and alveolar levels of IL-1β and S100A12 were measured in mice after acid-induced lung injury, with or without administration of RAGE inhibitors. Alveolar macrophages from patients with acute respiratory distress syndrome and from mechanically ventilated controls were analyzed using fluorescence-activated cell sorting. In vitro, RAGE promoted cytokine release and ROS production in macrophages and upregulated NLRP3 and TXNIP mRNA expression in response to S100A12. TXNIP inhibition downregulated NLRP3 gene expression and RAGE-mediated release of IL-1β by macrophages in vitro. In vivo, RAGE, NLRP3 and TXNIP lung expressions were upregulated during experimental acute lung injury, a phenomenon being reversed by RAGE inhibition. The numbers of cells expressing RAGE, NLRP3 and TXNIP among a specific subpopulation of CD16+CD14+CD206- (“pro-inflammatory”) alveolar macrophages were higher in patients with lung injury. This study provides a novel proof-of-concept of complex RAGE–TXNIP–NLRP3 interactions during macrophage activation in acute lung injury.

## 1. Introduction

Acute respiratory distress syndrome (ARDS) is a heterogeneous syndrome of diffuse alveolar injury leading to increased permeability, pulmonary edema, alveolar filling and rapid onset of hypoxemic respiratory failure [1,2]. Despite advances in our understanding of the pathogenesis of ARDS, no drug has reduced mortality in ARDS, and the syndrome is still under-recognized, undertreated and associated with high mortality [3,4]. There is growing evidence supporting a pivotal role for the receptor for advanced glycation end-products (RAGE) in the pathogenesis of ARDS [5,6,7,8]. RAGE is a transmembrane protein of the immunoglobulin superfamily that amplifies and perpetuates the inflammatory response [9,10,11]. RAGE is abundantly expressed in lung alveolar type (AT) 1 epithelial cells, but also in various other cell types, e.g., monocytes/macrophages, AT2 and endothelial cells. RAGE is constitutively expressed abundantly in the lung under basal conditions and its expression is enhanced during inflammatory states such as ARDS [12,13]. Its main soluble form, sRAGE, is secreted into the alveolar space and is detectable in the serum, serving as a biomarker for the degree of lung injury [7,12,13,14,15,16,17,18] and acting as a decoy receptor to downregulate the injurious pulmonary inflammatory response [19]. RAGE interacts with multiple ligands, including advanced glycation end-products (AGEs), high-mobility group box 1 protein (HMGB1), calgranulins/S100 proteins, amyloid peptides and macrophage adhesion ligand-1 (MAC-1) [20,21,22]. RAGE controls a variety of cellular processes such as cell proliferation and migration, inflammation, autophagy or apoptosis. Recent studies also point out a critical role for RAGE in modulating inflammation in macrophages [23], as cAMP elevation has been demonstrated to be capable of suppressing ligand-induced inflammatory reactions, including secretion of monocyte chemotactic protein 1 and macrophage recruitment, by promoting the conversion of RAGE isoforms from membrane RAGE to sRAGE [24].

The NOD-like receptor family, pyrin domain containing 3 (NLRP3) inflammasome is a multi-protein complex of the innate immune system expressed in myeloid cells, consisting of the NOD-like receptor NLRP3, the adaptor protein ASC (apoptosis-associated speck-like protein containing a CARD) and caspase 1 [25]. Whereas activation of the NLRP3 inflammasome can help host defense against invading bacteria and pathogens, excessive activation of the inflammasome can lead to inflammation-associated lung tissue injury [26]. Emerging studies suggest that NLRP3 inflammasome-driven activation in alveolar macrophages plays a critical role during ARDS [26,27,28]. NLRP3 inflammasome formation and activation require a canonical two-step mechanism involving the sequential ligand-dependent activation of toll-like receptors (e.g., TLR4) or related receptors that, in turn, stimulate NF-κB-dependent transcription of NLRP3 and pro-IL-1β, leading to NLRP3 and pro-IL-1β protein synthesis (priming) [29]. This is followed by a second signal generated by ion flux, phagosomal destabilization, mitochondrial reactive oxygen species (ROS) or mitochondrial damage associated molecular patterns (DAMPs), that results in NLRP3 inflammasome assembly and activation [30,31]. Once activated, caspase 1 proteolytically cleaves the cytokine precursors pro-IL-1β and pro-IL-18, resulting in the maturation and release of IL-1β and IL-18 [29]. As a pattern-recognition receptor, RAGE shares similarities with TLR4, including downstream activation of NF-κB-dependent [32] and ROS generation [33], with subsequent upregulation of RAGE expression itself [34]. In addition, in vitro studies suggest a role of RAGE in upregulated synthesis of pro-IL-1β and pro-IL-18 in THP-1 macrophages [35].

Recent findings demonstrate a potential role for thioredoxin-interacting protein (TXNIP) in innate immunity as a link between oxidative stress and NLRP3 inflammasome activation [36,37]. TXNIP was initially identified as one of the proteins that interacts with thioredoxin (TRX) and reduces its function of ROS scavenger, thus promoting oxidative stress [38]. TXNIP mediates inflammasome activation and injury in podocytes during diabetic nephropathy [39]. Furthermore, HMGB1 stimulates ROS production through TLR4 and a synergistic collaboration with RAGE signaling, in a mouse model of hemorrhagic shock [40]. In turn, ROS promote the association of TXNIP with NLRP3 and subsequent inflammasome activation in lung endothelial cells [40]. However, the role of TXNIP and RAGE-dependent mechanisms of TXNIP–NLRP3-driven macrophage activation are underinvestigated in the setting of acute lung injury [41,42,43].

In this study, we first tested the hypothesis that RAGE activation would induce NLRP3 activation and ROS production in a TXNIP-dependent manner. To test this hypothesis, we investigated the effects of S100A12–RAGE interaction on the production of IL-1β and ROS in cultured THP-1 macrophages. Next, we examined in vitro the role of TXNIP on IL-1β secretion and ROS production using small interfering (si)RNA knockdown. We further tested in vivo the hypothesis of RAGE–TXNIP–NLRP3-dependent mechanisms of lung injury using a mouse model of acid-induced lung injury and fluorescence-activated cell sorting (FACS) analysis of alveolar macrophages from patients with ARDS.

## 2. Results

### 2.1. S100A12 Stimulates ROS Production and IL-1β Secretion in a RAGE-Dependent Manner in THP-1 Cells

After stimulation by S100A12, cultured THP-1 macrophages released higher levels of cytokines IL-1β in the medium, compared to controls. S100A12-induced release of cytokines was inhibited, at least partially, by FPS-ZM1 (Figure 1). The production of intracellular ROS was increased by S100A12, a phenomenon inhibited by RAGE antagonist FPS-ZM1 (Figure 2).

### 2.2. Effects of siRNA-Targeted TXNIP and NLRP3 Knockdown on S100A12-Induced IL-1β Secretion and ROS Production by THP-1 Cells

siRNA-targeted knockdown of NLRP3 inhibited the S100A12-induced upregulation of IL-1β secretion by macrophages (Figure 3A). In contrast, siRNA-targeted TXNIP knockdown was associated with increased S100A12-induced IL-1β secretion into the medium, compared to TXNIP siRNA-transfected cells that were not exposed to S100A12 (Figure 3A).

NLRP3 and TXNIP knockdown THP-1 cells did not increase ROS production after exposure to S100A12, as compared to scramble siRNA-transfected S100A12-treated cells (Figure 3B).

### 2.3. RAGE-Dependent Regulation of Gene Expression in S100A12-Activated THP-1 Cells

RAGE mRNA expression was upregulated by S100A12, and this upregulation was inhibited by FPS-ZM1. This phenomenon was notably absent after TXNIP knockdown (Appendix A, online supplement).

mRNA expression levels of NLRP3 and TXNIP were both upregulated by S100A12 in THP-1 cells, in a RAGE-dependent manner (Appendix A, online supplement).

TXNIP knockdown macrophages had higher mRNA levels of NLRP3 than controls (Appendix A, online supplement). TXNIP siRNA knockdown was associated with increased mRNA levels of NLRP3, even in the absence of S100A12 (Appendix A, online supplement).

### 2.4. Anti-RAGE mAb and Recombinant sRAGE Prevent HCl Injury-Induced Upregulation of Lung NLRP3 and TXNIP In Vivo

In lungs from mice with acid-induced lung injury, NLRP3 and TXNIP protein expression was significantly upregulated on days 1–2, compared to sham animals, whereas lung mRNA levels of NLRP3 and TXNIP rose earlier after injury (from day 0) (Figure 4). Upregulated lung expression of NLRP3 and TXNIP after lung injury was alleviated by treatment with anti-RAGE mAB or sRAGE (Figure 4).

### 2.5. Effects of Acid-Induced Injury and RAGE Modulation on Alveolar Levels of IL-1β

Acid-induced injury significantly upregulated bronchoalveolar lavage (BAL) levels of IL-1β from day 0 to day 2, compared to sham animals, whereas treatment with sRAGE or anti-RAGE mAb significantly prevented the rise in BAL IL-1β on days 1–2 (Figure 5).

### 2.6. RAGE, TXNIP and NLRP3 Expression by Human Alveolar Macrophages during ARDS

The baseline characteristics and clinical outcomes of patients with or without ARDS are reported in the Table 1. Patients with ARDS had higher BAL levels of IL-1β and S100A12 than mechanically ventilated patients without ARDS (Figure 6).

In FACS analysis, groups did not significantly differ with regards to: the proportions of CD16+CD14- or CD16+CD14+ cells within the CD45+ population; the percentages of CD163+ and CD206+ cells within the CD16+CD14- subpopulation; the percentages of CD163+ cells within the CD16+CD14+ subpopulation; or the percentages of RAGE+, NLRP3+ and TXNIP+ cells within these latter subpopulations (Appendix A, online supplement). However, the percentage of CD206- cells within the CD16+CD14+ subpopulation was higher in patients with ARDS, compared to those without ARDS (Figure 7). In the CD16+CD14+CD206+ subpopulation, cells were more frequently RAGE+ in patients with ARDS than in those without (Appendix A, online supplement). In the CD16+CD14+CD206- subpopulation, there were more RAGE+ and NLRP3+ cells, but fewer TXNIP+ cells, in patients with ARDS than in those without ARDS (Figure 7).

## 3. Discussion

In this study, we found that: (1) S100A12–RAGE interaction mediates cytokine release and ROS production in THP-1 macrophages, through an upregulation of mRNA expression levels of NLRP3 and TXNIP; (2) TXNIP has a role in RAGE-mediated NLRP3 activation in vitro; (3) RAGE, NLRP3 and TXNIP expressions were increased after acute lung injury in a mouse model of ARDS, which was reversed by RAGE inhibition; (4) within pro-inflammatory (“M1-like”) CD16+CD14+CD206- alveolar macrophages, cells were more frequently RAGE+ and NLRP3+ cells, and less frequently TXNIP+, in patients with ARDS, compared to those without ARDS.

These results are in line with recent evidence of a prominent role of the NLRP3 inflammasome in ARDS [26,27,44], with increased BAL IL-1β and upregulated lung expression of NLRP3 in vivo after acid-induced lung injury. NLRP3-deficient (Nlrp3^−/−^) mice displayed increased mortality under hyperoxic conditions relative to wild-type and IL-1β^−/−^ mice. Under hyperoxia, Nlrp3^−/−^ mice displayed reduced lung inflammatory responses, including inflammatory cell infiltration and cytokine expression, but without differences in lung IL-1β production compared to wild-type mice [45]. Using a murine model of ventilator-induced lung injury (VILI), Dolinay et al. demonstrated that VILI upregulated several inflammasome-related genes including IL-1α, caspase-activator domain-10, IL-1 receptor and PYCARD (encoding for ASC protein, the inflammasome adaptor); in this model, genetic deletion of IL-18 or caspase 1 reduced lung injury, inflammation, alveolar-capillary barrier dysfunction and edema in response to injurious ventilation [46]. In human studies using plasma specimens from critically ill patients, the expression of inflammasome-related caspase 1, IL-18 and IL-1β mRNA transcripts was significantly higher in patients with sepsis or ARDS, compared with patients with systemic inflammatory response syndrome alone [46]. Importantly, circulating IL-18 and mitochondrial DNA, a DAMP associated with NLRP3 inflammasome activation, were significantly elevated in plasma from human patients critically ill with diseases such as sepsis and sepsis-induced ARDS, and significantly associated with ICU mortality [46,47]. Although previous studies underscored a role for inflammasome system and its regulated cytokines in the propagation of experimental ARDS, we report here for the first time that lung-injury-induced NLRP3 activation may be, at least in part, mediated by RAGE. Indeed, anti-RAGE therapy decreased BAL levels of IL-1β and prevented upregulation of mRNA and protein lung expression of NLRP3 in injured mice. Such findings are further supported by RAGE-dependent mechanisms of NLRP3 activation in cultured macrophages, as S100A12–RAGE interaction stimulates ROS production and IL-1β secretion in vitro, a phenomenon that was abrogated by NLRP3-targeted siRNA knockdown.

Our findings also provide novel insights into a role for TXNIP on RAGE-mediated NLRP3 activation in macrophages. Using specific siRNA knockdown in THP-1 cells, we could demonstrate that both TXNIP and NLRP3, which are upregulated in vivo during lung injury, seem mandatory for RAGE-induced upregulation of ROS production. The release of IL-1β by THP-1 macrophages necessitated NLRP3, but Il-1β levels were upregulated in the culture medium of TXNIP siRNA-transfected cells exposed to S100A12 in our study [48]. Such findings are surprising, because TXNIP has been previously described as a NLRP3 activator in the presence of ROS, with subsequent cytokine secretion [36,37,49,50]. Such a mechanism has been reported in podocytes and in Schwann cells, with downregulated IL-1β release after TXNIP knockdown [51,52]. Discrepancies with our findings may be explained by distinct responses in various cell types, various effects of distinct RAGE ligands (through RAGE itself, but also through other membrane receptors, including TLR4) or adaptive mechanisms that may have developed in macrophages from TXNIP knockout mice in previous studies. Nevertheless, because NLRP3 gene expression was increased in the absence of TXNIP in vitro, and because it was further increased by S100A12, we hypothesize that TXNIP may downregulate NLRP3 gene expression (even under basal culture conditions) and S100A12-RAGE-mediated release of IL-1β in macrophages. Interestingly, upregulated lung mRNA expression of TXNIP was, in turn, downregulated by treatment with anti-RAGE mAb or recombinant sRAGE in lung-injured mice, thus further supporting crosstalk or feedback mechanisms between TXNIP and RAGE pathways.

Our FACS analysis found significant differences in the number of cells expressing RAGE, NLRP3 and TXNIP among classically activated (M1)—or “proinflammatory”—human alveolar macrophages, but not among other tested subpopulations. Indeed, more CD16+CD14+CD206- cells expressed RAGE and NLRP3, and fewer CD16+CD14+CD206- cells expressed TXNIP, in patients with ARDS than in those without ARDS.

According to our findings from in vitro and animal studies, this could suggest that, at least in some M1-like macrophages, the activation of RAGE and NLRP3 pathways might be associated with a downregulation of TXNIP activity during ARDS. Whether interactions between RAGE, TXNIP and NLRP3 contribute to the presence and/or evolution of distinct alveolar macrophage subsets that regulate the inflammatory response and its resolution remains unknown [53]. Understanding the precise regulatory mechanisms for such a complex network deserves further investigation.

Our study has some limitations. First, the exploratory design of our study does not provide precise regulatory mechanisms linking RAGE, TXNIP and NLRP3 pathways. However, we report hypothesis-generating findings, and the first demonstration of a RAGE-mediated pathway of NLRP3 inflammasome activation in cultured macrophages and in an in vivo model of acute lung injury. In addition, we also explored for the first time the expressions of RAGE, NLRP3 and TXNIP in human alveolar macrophages, along with the in vivo effects of RAGE inhibition on TXNIP–NLRP3 activation during acute lung injury. Such novel findings have the potential to stimulate future research on precise intracellular pathways that interact upon RAGE, NLRP3 and TXNIP activation, along with their downstream targets, using additional siRNA targets, knockout animals and/or specific pharmacological antagonists of downstream effectors. Second, we only described the role of RAGE activation by its ligand S100A12 in THP-1 cells. As RAGE downstream pathways vary among cell types and depend on RAGE ligands, our results cannot be generalizable to other settings [9,33]. Third, we did not use primary cultures of macrophages in our study, and our findings may have been different in cells that may better reflect in vivo mechanisms. Fourth, the role of intercellular crosstalks within the lung (e.g., between macrophages and epithelial/endothelial cells), mitochondrial dysfunction and migratory properties of macrophages may significantly influence highly regulated cascades such as RAGE, TXNIP and NLRP3 pathways. Such explorations were out of the scope of our study but will be of great importance to better understand RAGE–NLRP3-driven alveolar inflammation and its role in ARDS. Finally, the choice of our ARDS animal model, based on direct lung injury by HCl, hampers extrapolability of our results to other causes of experimental ARDS.

## 4. Materials and Methods

### 4.1. Cell Experiments

Human THP-1 cells were obtained from the American Type Culture Collection and cultured in suspension in RPMI-Glutamax (Gibco) supplemented with streptomycin (100 μg·mL^−1^), penicillin G (100 UI·mL^−1^), amphotericin B (2.5 μg·mL^−1^) and 10% decomplemented fetal bovine serum (HyClone Standard, GE). THP-1 were differentiated into macrophages following treatment with 200 nM phorbol 12-myristate 13-acetate (Abcam) for 72 h [54]. Then, cells were treated for 24 h in the absence or presence of RAGE agonist S100A12 (40 nM, MyBiosource) [55,56], alone or in combination with RAGE antagonist FPS-ZM1 (1 µM, EMD Millipore) [57].

Supernatants were collected and stored at −20 °C before duplicate measurement of IL-1β (R&D Systems) using magnetic Luminex assay kits. Total RNA was extracted and gene expressions of RAGE (mRNA Refseq NM_001136), NLRP3 (NM_183395) and TXNIP (NM_006472) were assessed using semi-quantitative real-time polymerase chain reaction (human RT² Profiler™ PCR Array, Qiagen). For quantification of intracellular reactive oxygen species (ROS) production, the CellROX^®^ Deep Red Reagent kit was applied (Molecular Probes^®^).

For siRNA knockdown experiments, THP-1 cells were grown in 6-well plates and transfected with specific siRNA oligonucleotides for mouse TXNIP, NLRP3 (Dharmacon) or or with scrambled oligonucleotide (0.1 μM; siGENOME non-targeting siRNA #1, Dharmacon) using Lipofectamine RNAiMax transfection reagent (Invitrogen). Cells were harvested 48 h after transfection.

### 4.2. Animal Studies

Mouse experiments were approved by the Animal Ethics Committee of the French Ministère de l’Education Nationale, de l’Enseignement Supérieur et de la Recherche (approval number CE 67–12; 4 December 2012) and performed in accordance with relevant guidelines and regulations for animal experimentation, including the 3R principles.

C57BL/6JRj mice were anesthetized prior to orotracheal instillation of hydrochloric acid [7,58]. After a recovery period under humidified oxygen, mice were transferred to stabulation. Injured and sham animals were evaluated at baseline and at specified time-points (1, 2, 4 days) after acid instillation in injured mice [7,58,59], using a modification of previous in situ models [7,58,60,61]. After 30-min ventilation (tidal volume = 8 mL·kg^−1^, positive end-expiratory pressure = 6 cmH_2_O, respiratory rate = 160 breaths·min^−1^ and FiO_2_ = 1), BAL was sampled as previously described [7] and levels of BAL IL-1 (R&D Systems) were measured in duplicate using mouse magnetic Luminex assay kits (eBioscience).

To examine the effect of RAGE inhibition on acid-induced lung injury, lung-injured mice were divided into three groups. Acid-injured mice (HCl group) received intratracheal instillation of HCl 0.1M pH 1 (75 μL/mouse). The mAb group was injected intravenously with anti-RAGE monoclonal antibody (15 mg·kg^−1^, R&D Systems) 30 minutes before the HCl instillation [62,63], and the sRAGE group was administered recombinant mouse sRAGE (100 μg/mouse, R&D Systems) intraperitoneally 1 h after the HCl instillation [63,64].

The animals were sacrificed after the ventilation period and subjected to lung sampling for assessment of protein and mRNA expression levels. Lung proteins were extracted from right lung tissue homogenates and further analyzed by enzyme-linked immunoassays using mouse NLRP3 and TXNIP (MyBiosource) kits. In parallel, total RNA was isolated from left lung and gene expressions of RAGE (mRNA Refseq NM_007425), NLRP3 (NM_145827) and TXNIP (NM_011324) were assessed using semi-quantitative real-time polymerase chain reaction (mouse RT² Profiler™ PCR Array, Qiagen).

### 4.3. Human Study

Protocols were approved by the Institutional Review Board of the University Hospital of Clermont-Ferrand, France (Comité de Protection des Personnes Sud Est VI, approval number AU1151). All participants, or their next-of-kin, provided written consent to participate. The study was registered on www.clinicaltrials.gov (ClinicalTrials.gov Identifier: NCT02545621, accessed on 30 September 2022).

Five adult patients admitted to Clermont-Ferrand university hospital intensive care unit (ICU) within the first 24 h after onset of moderate to severe ARDS according to the 2012 Berlin definition (ARDS group) [65] and five age- and sex-matched ICU patients without ARDS but under mechanical ventilation for less than 24 h (Control group) were included in this monocenter prospective observational study. Exclusion criteria included: pregnancy, acute exacerbation of diabetes (ketoacidosis, hyperosmolar hyperglycemic state), dialysis-dependent chronic renal failure, Alzheimer’s disease, amyloidosis, evolutive neoplastic lesion, chronic pulmonary disease requiring long-term oxygen therapy or mechanical ventilation, chemotherapy treatment in the last 30 days, severe neutropenia (<0.5 G/L).

At inclusion, fiberoptic-guided BAL (performed with 2 × 50 mL NaCl 0.9%) was analyzed using fluorescence-activated cell sorting (FACS). Alveolar macrophages were isolated and stained depending on their expression of membrane markers of M1-like (“pro-inflammatory”) or M2-like (“anti-inflammatory”) phenotypes [66], membrane RAGE and intracellular TXNIP and NLRP3. The following antibodies were used: rat anti-human NLRP3 monoclonal antibody (MAB7578, R&D Systems), goat anti-human C18-conjugated VDUP1/TXNIP polyclonal antibody (SC-33099, Santa Cruz), rabbit anti-human APC-conjugated anti-RAGER polyclonal antibody (LS-C212626, Lifespan Biosciences), APC-H7 mouse anti-human CD14, V450 mouse anti-human CD16, V500 mouse anti-human CD45, BV711 mouse anti-human CD163 and PE-CF594 mouse anti-human CD206 antibodies (BD Biosciences). Supernatants were analyzed for levels of S100A12 (CycLex Co) and IL-1β (R&D Systems) using duplicate ELISA.

### 4.4. Statistical Analysis

Categorical data were expressed as numbers and percentages, and quantitative data as mean ± standard deviation (SD) or median and interquartile range (IQR) according to statistical distribution. For studies of protein and mRNA expression, data are presented as ratio to expression in control cell conditions or in sham animals. Statistical analyses were performed using Kruskal–Wallis with Bonferroni tests for pairwise comparisons between each time-point and sham controls (represented as day 0), or between various cell culture conditions. A limited number of animals was used for baseline comparisons (*n* = 3–4), and 4–6 animals were used in each group on days 1, 2 and 4. Comparisons of patients characteristics between groups were performed using the chi-squared or Fisher’s exact tests for categorical variables, and Student t-test or Mann–Whitney test were used when assumption if t-test were not met (normality and homoscedasticity studied using Fisher–Snedecor test) for quantitative variables. Analyses were performed using Prism 6 (Graphpad Software, La Jolla, CA, USA) and Stata software (version 13, StataCorp, College Station, TX, USA). A *p* < 0.05 (two-sided) was considered significant.

## 5. Conclusions

In this study, S100A12–RAGE interaction mediated cytokine release and ROS production by macrophages in vitro. TXNIP had a crucial role in S100A12–RAGE-mediated NLRP3 activation and production of IL-1β and of intracellular ROS by macrophages (Figure 8). RAGE, NLRP3 and TXNIP expressions were increased in an animal model of acute lung injury, a phenomenon that was reversed by RAGE inhibition. Proinflammatory CD16+CD14+CD206- alveolar macrophages expressing RAGE and NLRP3 were more frequent, and those expressing TXNIP less frequent, in patients with ARDS than in those without ARDS, supporting a novel proof-of-concept of RAGE–TXNIP–NLRP3-driven mechanisms of macrophage activation during acute lung injury.

## Figures and Tables

**Figure 1 ijms-23-11659-f001:**
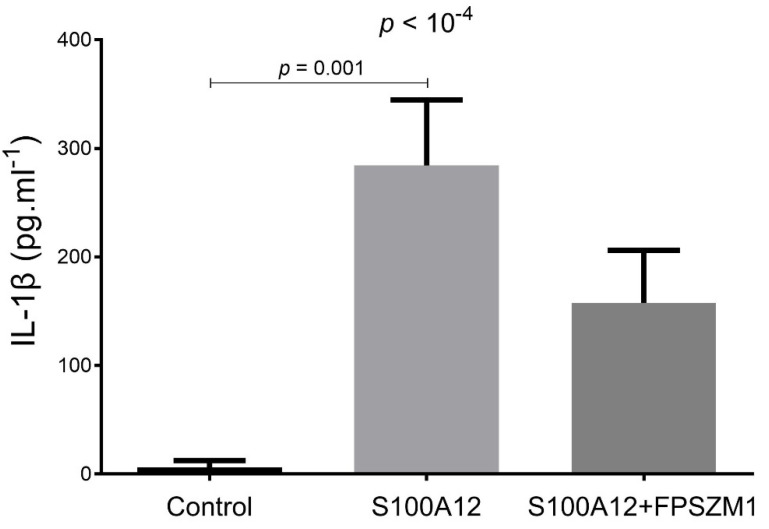
Levels of IL-1β in the culture medium of THP-1 cells treated for 24 h in the absence/presence of RAGE agonist S100A12 (40 nM, MyBiosource) alone or in combination with RAGE antagonist FPS-ZM1 (1 µM, EMD Millipore). Supernatants were collected and stored at −20 °C before duplicate measurement of IL-1β (R&D Systems) using magnetic Luminex assay kits.

**Figure 2 ijms-23-11659-f002:**
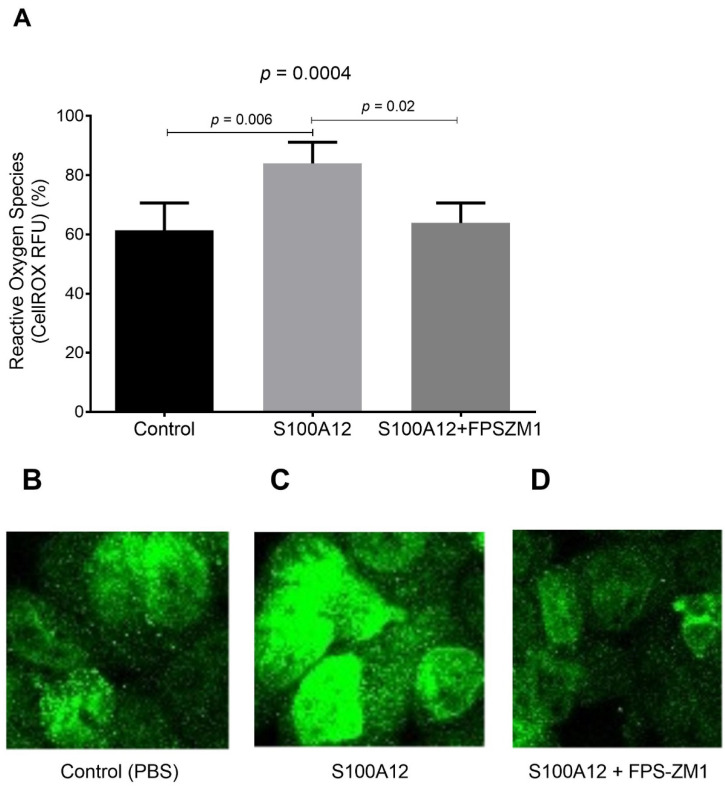
(**A**) Quantitative analysis of ROS measurements using CellROX Deep Red Reagent kit (Molecular Probes) in THP-1 cells treated for 24 h in the absence/presence of RAGE agonist S100A12 (40 nM, MyBiosource) alone or in combination with RAGE antagonist FPS-ZM1 (1 µM, EMD Millipore). Data are expressed as relative fluorescent units (RFU, in %). Representative images of CellROX staining in THP-1 cells treated with (**B**) PBS (control), (**C**) S100A12 alone or (**D**) S100A12 in combination with FPS-ZM1. Images of ROS staining were photographed under a microscope (Zeiss Axiophot). Image processing was conducted by Image J software 1.49.

**Figure 3 ijms-23-11659-f003:**
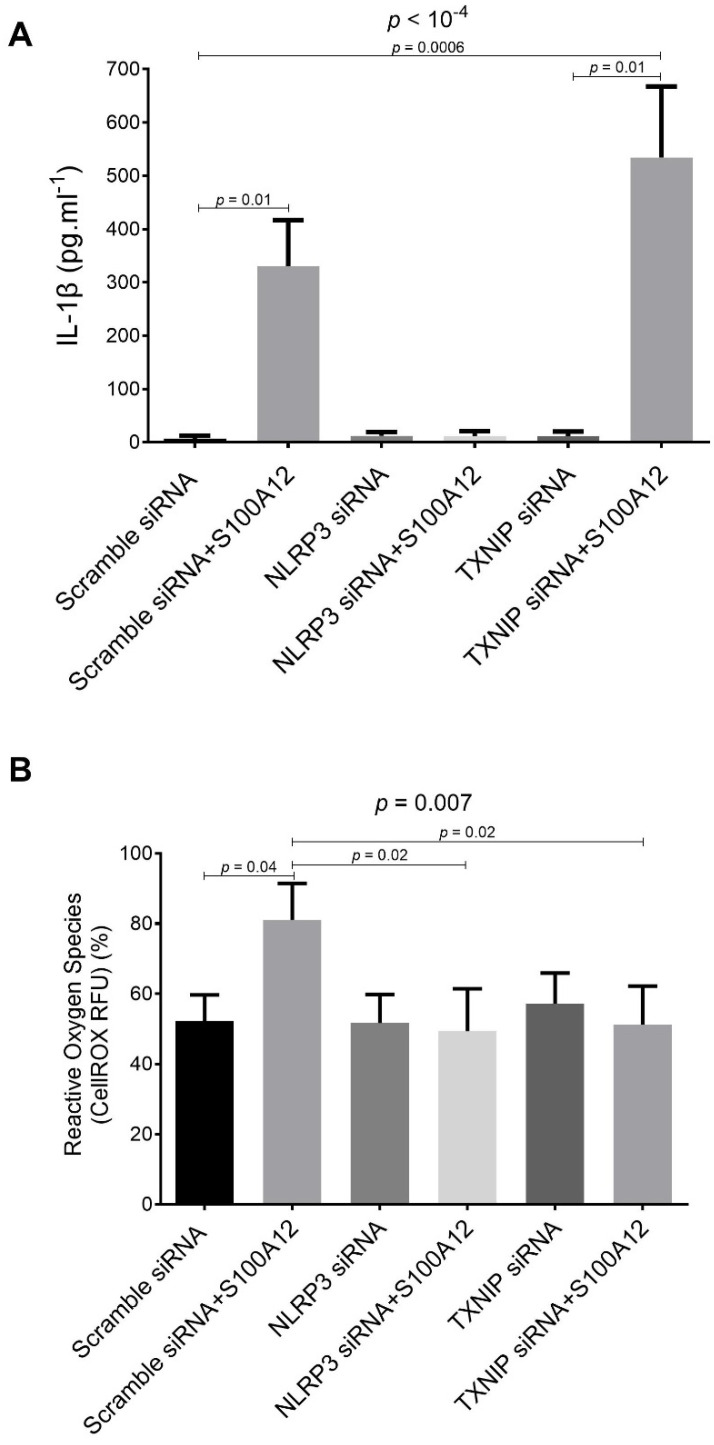
(**A**) Levels of IL-1β in the culture medium of THP-1 cells treated for 24 h in the absence/presence of RAGE agonist S100A12 (40 nM, MyBiosource) alone or in combination with RAGE antagonist FPS-ZM1 (1 µM, EMD Millipore). Supernatants were collected and stored at −20 °C before duplicate measurement using magnetic Luminex assay kits (R&D Systems). (**B**) Quantitative analysis of intracellular ROS production using CellROX Deep Red Reagent kit (Molecular Probes) in THP-1 macrophages transfected with specific siRNA oligonucleotides for mouse TXNIP (siRNA TXNIP, Dharmacon), NLRP3 (siRNA NLRP3, Dharmacon) or with scrambled oligonucleotide (0.1 μM Scramble siRNA; Dharmacon). Efficiency of siRNA knockdown was verified with duplicate ELISA of TXNIP (CycLex Co) and NLRP3 (Antibodies-online GmbH). Cells were treated for 24 h in the absence/presence of RAGE agonist S100A12 (40 nM, MyBiosource) alone or in combination with RAGE antagonist FPS-ZM1 (1 µM, EMD Millipore). ROS production is expressed in relative fluorescent units (RFU, in %).

**Figure 4 ijms-23-11659-f004:**
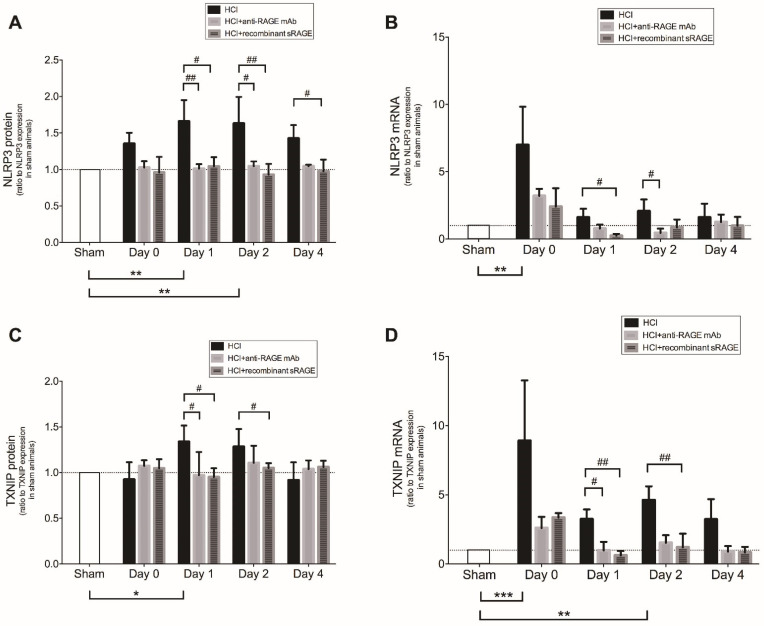
Lung levels of NLRP3 (**A**) protein and (**B**) mRNA, TXNIP (**C**) protein and (**D**) mRNA in acid-injured mice (HCl group), acid-injured mice treated with anti-RAGE monoclonal antibody (HCl+anti-RAGE mAb group) or with recombinant sRAGE (HCl+sRAGE group) and in uninjured, untreated mice (Sham group) (*n* = 4–6 for each time point). Threshold levels of mRNA expression (∆∆Ct) were normalized to housekeeping genes. Protein and mRNA expression levels are expressed as ratios to those in sham animals. Values are reported as means ± SD and are analyzed with Kruskal–Wallis test (nonparametric data). * *p* < 0.05, ** *p* < 10^−2^, *** *p* < 10^−3^ versus sham animals. # *p* < 0.05, ## *p* < 10^−2^ versus acid-injured mice.

**Figure 5 ijms-23-11659-f005:**
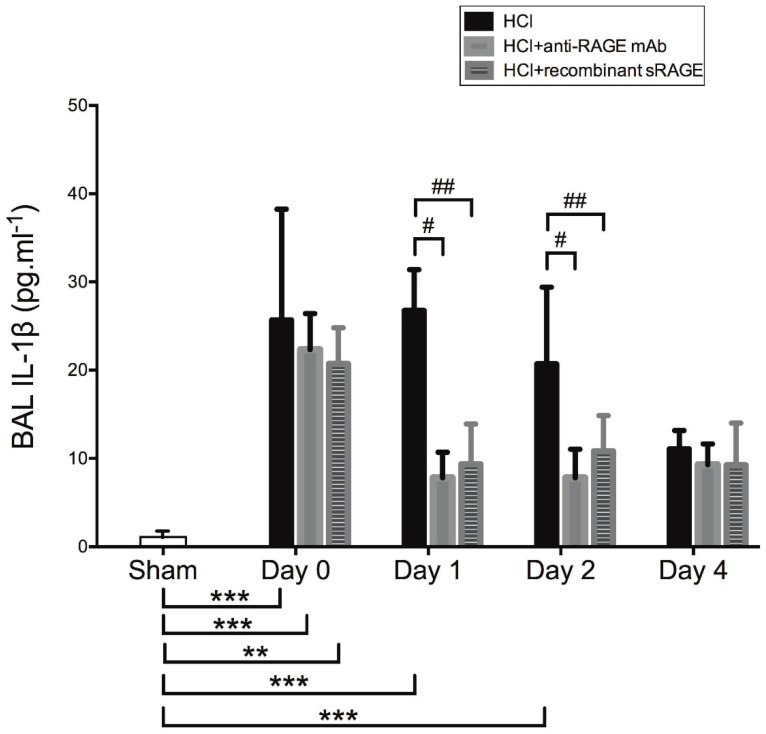
Measurement of bronchoalveolar lavage (BAL) levels of interleukin (IL)-1β in acid-injured mice (HCl group), acid-injured mice treated with anti-RAGE monoclonal antibody (HCl+anti-RAGE mAb group) or with recombinant sRAGE (HCl+sRAGE group) and in uninjured, untreated mice (Sham group) (*n* = 4–6 for each time point). Values are reported as means ± SD and are analyzed with Kruskal–Wallis test (nonparametric data). ** *p* < 10^−2^, *** *p* < 10^−3^ versus sham animals. # *p* < 0.05, ## *p* < 10^−2^ versus acid-injured mice.

**Figure 6 ijms-23-11659-f006:**
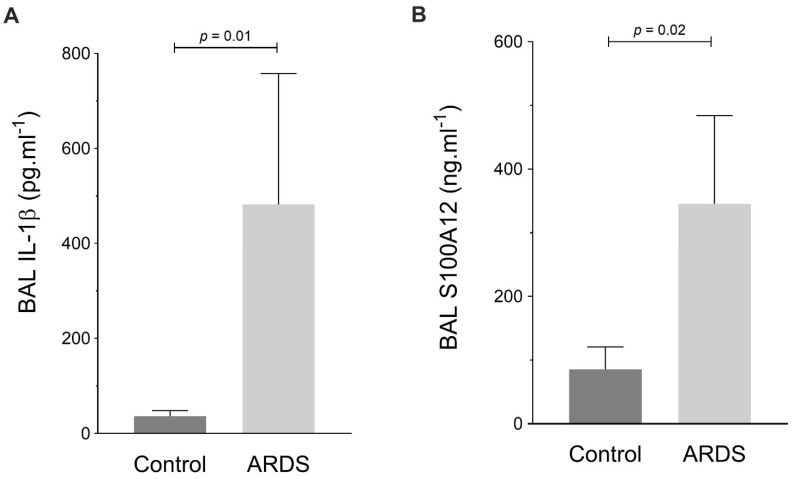
Measurement of bronchoalveolar lavage (BAL) levels of (**A**) interleukin (IL)-1β (in pg·mL^−1^) and (**B**) S100A12 (in ng·mL^−1^) in patients with acute respiratory distress syndrome (ARDS) and in mechanically ventilated patients without ARDS (Control). Values are reported as medians ± interquartile ranges and are analyzed with the Mann–Whitney test (nonparametric data).

**Figure 7 ijms-23-11659-f007:**
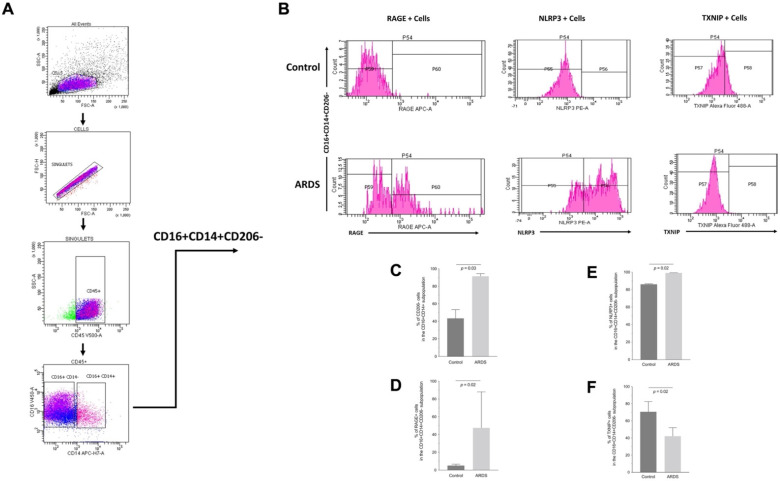
(**A**) Flow cytometry gating strategy used to identify pro-inflammatory (“M1-like”) CD16+CD14+CD206- alveolar macrophages. (**B**) Representative FACS images for the cell-surface expression of RAGE, NLRP3 and TXNIP. C-F) Quantification of FACS images: percentages of CD206- cells within the CD16+CD14+ subpopulation (**C**), RAGE+ cells within the CD16+CD14+CD206- subpopulation (**D**), NLRP3+ cells within the CD16+CD14+CD206- subpopulation (**E**) and of TXNIP+ cells within the CD16+CD14+CD206- subpopulation (**F**) of alveolar macrophages from patients with acute respiratory distress syndrome (ARDS) and in mechanically ventilated patients without ARDS (Control). Values are reported as medians ± interquartile ranges and are analyzed with the Mann–Whitney test (nonparametric data).

**Figure 8 ijms-23-11659-f008:**
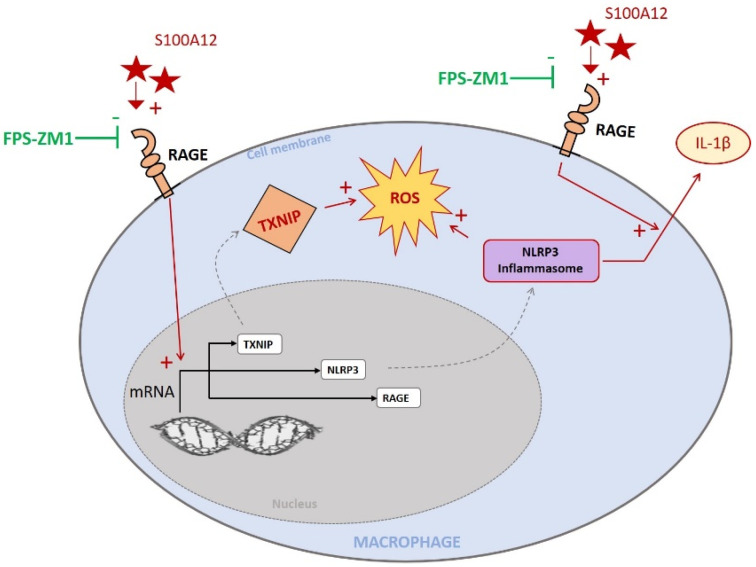
RAGE ligand S100A12 stimulates IL-1β release, ROS production and increased TXNIP, NLRP3 and RAGE mRNA levels. These effects were inhibited by RAGE antagonist FPS-ZM1. We demonstrated that TXNIP has a crucial role in S100A12–RAGE-mediated NLRP3 activation and production of IL-1β and of intracellular ROS by macrophages. In addition, RAGE, NLRP3 and TXNIP were increased in an animal model of acute lung injury and in proinflammatory CD16+CD14+CD206- human alveolar macrophages.

**Table 1 ijms-23-11659-t001:** Baseline Characteristics and Clinical Outcomes of Patients without or with acute respiratory distress syndrome (ARDS).

Characteristic	Control (n = 5)	ARDS (n = 5)	*p*-Value
*Demographics*			
Male sex, n (%)	3 (60)	3 (60)	1.0
Age, years	61 ± 11	69 ± 3	0.5
Body mass index, kg·m^−2^	28 ± 1	26 ± 1	0.4
*Coexisting Conditions, n (%)*			
Arterial hypertension	1 (20)	1 (20)	1.0
Type 2 diabetes	1 (20)	1 (20)	1.0
Chronic obstructive pulmonary disease	0 (0)	0 (0)	--
Current smoking	1 (20)	1 (20)	1.0
Chronic kidney disease	1 (20)	0 (0)	0.3
Hematologic neoplasm	0 (0)	0 (0)	--
Solid cancer	0 (0)	2 (40)	0.1
*Indication for ICU Admission, n (%)*			0.4
Septic shock of digestive origin	0 (0)	1 (20)	
Hemorrhagic shock	3 (0)	0 (0)	
Coma (intoxication)	1 (20)	0 (0)	
Acute respiratory failure	0 (0)	4 (80)	
Status epilepticus	1 (20)	0 (0)	
*Baseline Respiratory Variables*			
PEEP, cmH_2_O	8 [7–8]	14 [10–14]	0.02
Tidal volume, mL·kg^−1^ PBW	6.9 [6.6–7.7]	6.2 [5.9–6.6]	0.06
PaO_2_/FiO_2_, mmHg	164 [162–333]	166 [153–167]	0.6
PaCO_2_ mmHg	34 [31–37]	43 [40–53]	0.06
Inspiratory plateau pressure, cmH_2_O	15 [13–16]	29 [24–30]	0.04
*Baseline Hemodynamic Status*			
Mean arterial blood pressure, mmHg	89 ± 18	82 ± 9	0.6
Heart rate, per min	97 ± 23	97 ± 14	0.8
Need for norepinephrine, n (%)	1 (20)	3 (75)	0.2
Serum creatinine, μmol·L^−1^	87 ± 53	118 ± 69	0.5
Sequential Organ Failure Assessment score	6.6 ± 2.9	8.2 ± 3.8	0.4
*Clinical Outcomes*			0.5
Death at day 28, n (%)	1 (20)	3 (60)	
Ventilator-free days at day 28	25 [24–27]	0 [0–21]	

Data are presented as means ± standard deviations (SD), medians [interquartile ranges] or numbers with percentages. *p*-values were calculated for comparisons between patients without ARDS (control) and those with ARDS. ICU: intensive care unit. PEEP: positive end-expiratory pressure. PaO_2_: partial pressure of arterial oxygen. FiO_2_: fraction of inspired oxygen.

## Data Availability

The data presented in this study will be available on request from the corresponding author after publication of this article. Data transfer agreements may be required.

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
