# Peer review of "Receptor for Advanced Glycation End-Products Promotes Activation of Alveolar Macrophages through the NLRP3 Inflammasome/TXNIP Axis in Acute Lung Injury"

_ijms, 2022, doi:10.3390/ijms231911659_

Round 1

Reviewer 1 Report

There are several deficits in the manuscript. Authors should experimentally address these review comments

.

1. The title of this manuscript was not clear and could not fully support the conclusion of this study. Please revise it.

2. Time- and concentration-dependent effect of S100A12, FPS-ZM1 and siRNAs should be shown.

3. The protein expression of NLRP3 and TXNIP should be examined by western blot analysis both in in vitro and in vivo studies.

4. The ROS images in THP-1 macrophages should be shown.

5. The results of histological examination and injury score for acute lung injury should be shown.

6. The sample size of clinical study was too small to support the hypothesis of this study.

7. The results (images) of FACS analysis should be shown.

8. For the convenience of readers, the graphic abstract of this study is required.

Reviewer 2 Report

This is a well-planned study looking at AGE/RAGE and NLRP3-inflammasome axis in acute lung injury via TXNIP modulation. S100A12, a RAGE agonist was used as a stimulus, and changes in ROS levels as well as cytokine release were determined. The study results from the in vitro studies were corroborated in vivo animal and patient samples is a strength of this paper. The animal models used are appropriate to test the hypothesis. My only comment would be to justify the choice of the animal model and the stimulus (S100A12) over HMGB1 or some other RAGE agonist.

Round 2

Reviewer 1 Report

1. How authors got the  quantitative results of marker expression including RAGE, NLRP3, and TXNIP without the images of FACS analysis?

2. The images of FACS analysis should be incorporated into this manuscript.

3. How authors ensure the success in the establishment of acute lung injury model without histological examination?

Round 3

Reviewer 1 Report

I have no further comment.